# Light triggers a network switch between circadian morning and evening oscillators controlling behaviour during daily temperature cycles

**Clara Lorber, Solene Leleux, Ralf Stanewsky**[ID]**\*, Angelique Lamaze\***

Institute of Neuro and Behavioral Biology, Westfälische Wilhelms University, Münster, Germany

\* stanewsky@uni-muenster.de (RS); angie0203@hotmail.com (AL)

**Data Availability Statement:** All relevant data are within the manuscript and its Supporting Information files.

## Abstract

Proper timing of rhythmic locomotor behavior is the consequence of integrating environmental conditions and internal time dictated by the circadian clock. Rhythmic environmental input like daily light and temperature changes (called Zeitgeber) reset the molecular clock and entrain it to the environmental time zone the organism lives in. Furthermore, depending on the absolute temperature or light intensity, flies exhibit their main locomotor activity at different times of day, i.e., environmental input not only entrains the circadian clock but also determines the phase of a certain behavior. To understand how the brain clock can distinguish between (or integrate) an entraining Zeitgeber and environmental effects on activity phase, we attempted to entrain the clock with a Zeitgeber different from the environmental input used for phasing the behavior. 150 clock neurons in the *Drosophila melanogaster* brain control different aspects of the daily activity rhythms and are organized in various clusters. During regular 12 h light: 12 h dark cycles at constant mild temperature (LD 25˚C, LD being the Zeitgeber), so called morning oscillator (MO) neurons control the increase of locomotor activity just before lights-on, while evening oscillator (EO) neurons regulate the activity increase at the end of the day, a few hours before lights-off. Here, using 12 h: 12 h 25˚C:16˚C temperature cycles as Zeitgeber, we attempted to look at the impact of light on phasing locomotor behavior. While in constant light and 25˚C:16˚C temperature cycles (LLTC), flies show an unimodal locomotor activity peak in the evening, during the same temperature cycle, but in the absence of light (DDTC), the phase of the activity peak is shifted to the morning. Here, we show that the EO is necessary for synchronized behavior in LLTC but not for entraining the molecular clock of the other clock neuronal groups, while the MO controls synchronized morning activity in DDTC. Interestingly, our data suggest that the influence of the EO on the synchronization increases depending on the length of the photoperiod (constant light *vs* 12 h of light). Hence, our results show that effects of different environmental cues on clock entrainment and activity phase can be separated, allowing to decipher their integration by the circadian clock.

**Funding:** AL received a Women in Research (WiRe) (Women in Research) fellowship, co-funded by the University of Münster (https://www.uni-muenster.de/en/) and the Deutsche Forschungsgemeinschaft (DFG) (https://www.dfg.de/en/index.jsp). AL and RS are supported by the DFG grant STA-421/7-1. We acknowledge support from the Open Access Publishing Fund of University of Münster. The funders had no role in study design, data collection and analysis, decision to publish, or preparation of the manuscript.

**Competing interests:** The authors have declared that no competing interests exist.

## Author summary

"If a clock is to provide information involved in controlling important functions, then clearly it must be reasonably reliable" said Colin Pittendrigh, one of the chronobiology pioneers in 1954. The circadian clock allows organisms to synchronize with their ecological niche. For this, the circadian clock uses rhythmic environmental parameters (Zeitgeber), the main ones being light and temperature. Hence, Colin Pittendrigh posted a still unresolved enigma in chronobiology. How can a clock be reliable when its resetting depends on environmental fluctuations that are not so reliable? Both, light and temperature vary a lot on a day-to-day basis, and animals respond to these variations depending on the time of day.

Here, we propose a new model where the molecular clock resets to environmental cycles in a robust and independent manner, while the underlying neuronal oscillatory network switches its balance towards specific oscillators depending on the environmental condition thereby leading to distinct behavioral adaptation. To proof this proposed dogma in fruit flies, using temperature cycles as Zeitgeber, we demonstrate a light-induced switch of the network balance. Hence, we supply a foundation that in the future will help to understand how animals use their circadian clock to adapt their behavior to environmental changes.

## Introduction

An important function of the circadian clock is to maintain the synchronization of the organism with its ecological temporal niche in concert with the natural environmental fluctuations. Impaired clock synchronization not only negatively impacts fitness of animals but also leads to severe physical and mental syndromes in humans [1, 2]. To stay on time, the circadian clock is reset everyday by light changes as well as temperature oscillations, commonly referred to as 'Zeitgeber' (German for 'time giver'). Molecularly, circadian clocks are composed of a set of core clock genes, which regulate their own temporal expression in form of 24-h transcriptional negative feedback loops [3]. In *Drosophila* the transcription factors CLOCK (CLK) and CYCLE (CYC) form a heterodimer that promotes the expression of the *period* (*per*) and *timeless* (*tim*) genes. After cytoplasmic accumulation and several post-translational modifications, PER and TIM translocate to the nucleus to inhibit their own transcription by interacting with CLK/CYC. Eventually, PER and TIM are degraded, allowing CLK/CYC to start a new round of this ~24 h molecular cycle. The TIM protein is the link that allows for adjustment of these molecular oscillations to light:dark cycles. When activated by blue light, the circadian photoreceptor CRYPTOCHROME (CRY) interacts with TIM, and both proteins subsequently become degraded after interacting with the F-box ubiquitin ligase JETLAG (JET) [4–6]. However, CRY is not expressed in all the clock cells [7] and other CRY-independent mechanisms exist that mediate light- and activity-dependent degradation of TIM [8–10].

In 12h-12h light-dark and constant 25°C (LD) conditions, fruit flies exhibit crepuscular behavior and increase their locomotion twice a day: a couple of hours before light-on (morning anticipation) and about three hours before light-off (evening anticipation). These two loco-motor activity peaks are controlled by two different oscillators, which nevertheless show identical peaks and troughs of their molecular oscillations [11, 12]. The current view to explain this conundrum is that the neuronal activity of these two oscillators cycles with a different phase [13–15]. The *Drosophila* brain clock containing these neuronal oscillators is composed of about 150 neurons distributed along the lateral and dorsal part of the protocerebrum. They

all express the core clock genes, however they can be distinguished by their anatomical position, projection patterns, as well as the neurotransmitters and neuropeptides they express. The Pigment Dispersing Factor (PDF) neuropeptide is expressed in only eight clock neurons per hemisphere in the ventro-lateral anterior brain: four large ventro-lateral neurons (l-LNv), projecting into the accessory medulla and the contralateral optic lobe, as well as four small LNv (s-LNv), which form the so called morning oscillator (MO) and project into the ipsilateral dorsal protocerebrum. The evening oscillator (EO) is composed of three dorso-lateral neurons (LNd) and a 5th s-LNv that does not express PDF [11, 12]. Both MO and EO express CRY [16]. In addition, there are four groups of Dorsal Neurons (DN1a, DN1p, DN2, and DN3). The more posteriorly located DN1p are a heterogeneous group consisting of both CRY+ and CRY- neurons. While the CRY+ DN1p contribute to the control of morning activity, the CRY- neurons control evening activity, albeit under restrictive light conditions [17–19]. The two more anteriorly located DN1a are part of the EO and express CRY [11, 20]. Finally, three CRY- Lateral Posterior Neurons (LPN) may influence behavior during temperature cycles, based on preferential synchronization of LPN clock protein oscillations to temperature as compared to LD cycles [21–23]. Constant light stops the clock in all clock cells and as a consequence, flies are arrhythmic in this condition [24], presumably because the constant activity of CRY leads to constant degradation of TIM [5, 25].

Strikingly, temperature cycles are able to overcome LL-induced arrhythmicity in flies carrying a recently evolved allele of the *tim* gene, called *ls-tim* [22, 26, 27]. In contrast, flies bearing the original *s-tim* allele behave arrhythmic during temperature cycles in LL. This is, because the S-TIM protein (the only form of TIM encoded by *s-tim* flies) has a high affinity to CRY, whereas L-TIM (*ls-tim* flies generate both S-TIM and L-TIM) has strongly reduced affinity to CRY and is therefore more stable in the light [28]. Consequently, in the absence of CRY, *s-tim* flies also behave rhythmically in LL and temperature cycles [27]. In constant light and 12 h 25°C: 12 h 16°C (LLTC, 25°C-16°C), *ls-tim* flies are rhythmic with a single synchronized activity peak occurring in the evening [26, 27, 29], while in constant darkness the same temperature cycle (DDTC) leads to a behavioral activity peak in the morning in both *s-tim* and *ls-tim* flies [27]. Hence, while temperature is used as a Zeitgeber, the presence or absence of light influences the activity phase, presumably by modulating the balance toward one dominant neuronal oscillator. Therefore, we postulated that the EO is responsible for the evening output in LLTC, while the MO takes over in DDTC [30]. To test this, we genetically manipulated the different clock neuronal subsets, including the morning and evening oscillators, and analyzed the consequences on locomotor activity in LD, and in temperature cycles during constant light and constant darkness (LLTC and DDTC). We show that the evening oscillator operating in LD also regulates the evening peak in LLTC, while in DDTC morning oscillator neurons determine the phase of active locomotion. Hence, our results point to an environment-dependent switch (in this case the presence or absence of light) between different oscillators controlling daily activity phases of the fly.

## Results

### EO neurons require a functional circadian clock to synchronize evening activity during temperature cycles in constant light

Under standard LD conditions *Drosophila melanogaster* exhibits crepuscular behavior with periods of activity centered around the light transitions in the morning and evening. Interestingly, the evening activity peak observed in LD, is also observed in LLTC [29], albeit with a phase advance compared to LD. We hypothesized that the timing of this activity peak is controlled by the same clock neurons, known as the evening oscillator (EO). Restricting clock

function to these cells is sufficient to drive the evening peak in LD [11]. To test whether this oscillator is required for controlling the behavioral evening activity phase, we interfered with clock function in restricted groups of clock neurons using the dominant negative form of *cycle* (*UAS-cyc^{DN}*) [31] (Figs 1 and S1, Table 1). Both *Mai179-Gal4* and *cry-Gal4[19]-* are expressed in the evening oscillator but also in the s-LNv and the l-LNv [11, 30]. However, *Mai179-Gal4* is only weakly expressed in morning oscillator neurons [11]. Furthermore, while *cry-Gal4[19]* expression is restricted to clock neurons, *Mai179-Gal4* is broadly expressed in non-clock neurons. Flies were exposed to three days of LD followed by two to three days in LL, and subsequent LLTC, which was delayed by 5 h compared to LD (Fig 1A). Interestingly, morning anticipation in LD was not affected by the expression of *cyc^{DN}* in the s-LNv (Fig 1A–1C), presumably because clock function in the DN1p is sufficient [12, 18]. As expected, expression of *cyc^{DN}* in the morning and evening oscillator with both drivers strongly reduced the amplitude of the evening peak, which anticipates the environmental transition in both LD and LLTC (Fig 1A–1C). While the startle response to lights-off and the following rapid activity decline in LD was not affected, both *Mai179-Gal4* and *cry-Gal4[19] > cyc^{DN}* flies showed increased activity in the cryophase during LLTC (Fig 1A–1C). In the following we focus on the amplitude of the anticipatory behavioral evening peak, because it is under tight clock-neuronal control and also influenced by the environment. A decrease in amplitude can be due to reduced synchronization within the population, or to reduced speed of activity increase resulting in a delayed activity peak. To first quantify the amplitude reduction, we calculated the slope for each fly (the speed of activity increase, Δactivity/Δt). The two time points (start of activity increase and activity peak) were determined based on median activity levels (S1A Fig and Materials and Methods). Using this method, we observed a strong decrease of the slope in both *Mai179>cyc^{DN}* and *cry[19]>cyc^{DN}* flies in LD and LLTC (Figs 1D, 1E and S1B). To test whether this decrease was due to a decrease of synchronization, or a (synchronized) delay of anticipatory activity, we compared the median of the slope value (Slo_{exp}) with the slope of the median (Slo_{the}). Therefore, if fly activity is highly synchronized between individuals, the ratio of Slo_{exp}/Slo_{the} is close to 1 (i.e., most individuals behave similar to the median). While the Slo_{exp}/Slo_{the} ratio for the controls was close to 1, it was reduced to ~0.5 for both *Mai179>cyc^{DN}* and *cry[19]>cyc^{DN}* flies (S1B Fig). In addition, to easily visualize if a population is synchronized, we compared the percentage of flies with a Slo_{exp} > ½ x Slo_{the} with flies that are below this value. Indeed, > 80% of the control flies increase their locomotion with Slp_{exp} > ½ Slp_{the}, compared to only 50–60% of the *Mai179>cyc^{DN}* and *cry[19]>cyc^{DN}* flies (S1C Fig, blue bars percentage of flies with Slo_{exp} > ½ of Slo_{the} 'anticipating', orange bars percentage of flies with Slo_{exp} < ½ of Slo_{the} 'non-anticipating'). Taken together, these results indicate that expression *cyc^{DN}* in both evening and morning neurons reduces behavioral synchronization of evening activity in both LD and LLTC.

## MO and EO neurons are not required for temperature entrainment of other clock neurons

How temperature cycles entrain the molecular clock in the brain is not known. Previous observations suggest that temperature entrainment uses multiple molecular and probably neuronal circuits involving peripheral thermosensors [32–36]. We therefore tested, whether the absence of a functional molecular clock in the *Mai179* cells could disturb the other clock cells in the brain. For this, we dissected brains and quantified PER levels on the 6^{th} day of LLTC at four different time points. First, we confirmed the circadian clock disruption after *cyc^{DN}* expression in Mai179 cells, because we observed only 3 out of 6 LNd at ZT0 (Fig 2A), while PER levels were constitutively low in the other *Mai179* cells (Fig 2C). In the remaining CRY⁻ LNd, we

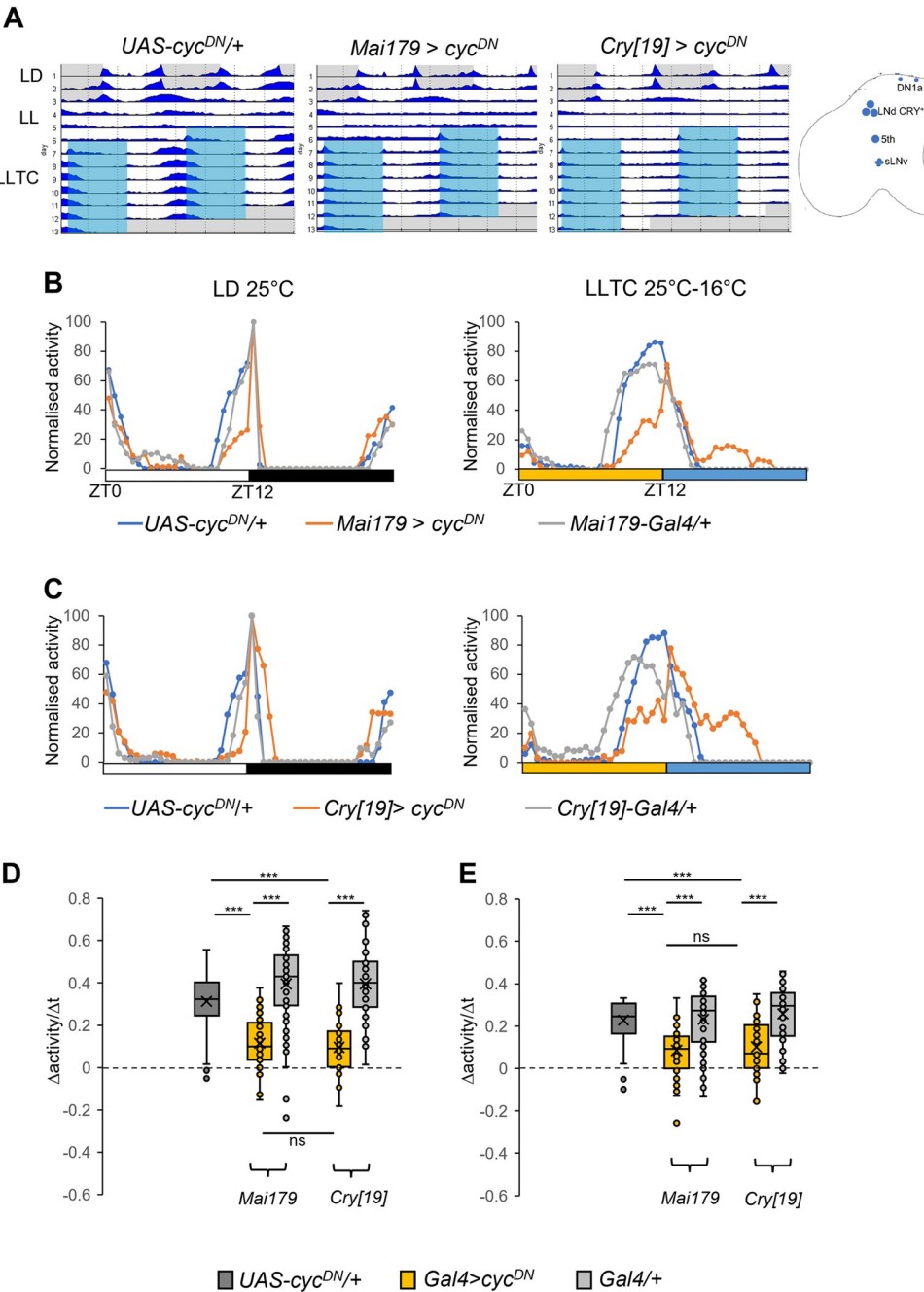

**Fig 1. MO and EO circadian clocks are required for synchronization to temperature cycles in constant light.** Male flies were synchronized to LD 25˚C for two days, before being exposed to LL 25˚C for three days, followed by temperature cycles in LL (LLTC 25˚C:16˚C), which were delayed by 5-h compared to the initial LD cycle. A) Double plotted average actograms of one representative experiment. N: $UAS\text{-}cyc^{DN}/+$ = 19, $Mai179>cyc^{DN}$ = 21, $cry[19]>cyc^{DN}$ = 22. Conditions are indicated to the left. White areas indicate lights on and 25˚C and grey areas lights off and 25˚C during LD, and blue areas lights-on and 16˚C during LLTC. Cartoon on the right shows clock neurons expressing $Mai179$ and $cry[19]$ drivers. B-C) Median of normalized locomotor activity during the last day of LD (left) and the 6th day of LLTC (right). White bars represent lights-on, black bar lights off in LD (left), yellow and blue bars indicate thermophase (25˚C) and cryophase (16˚C) in LL, respectively (right). N: $UAS\text{-}cyc^{DN}/+$ = 59, $Mai179>cyc^{DN}$ = 55, $Mai179\text{-}Gal4/+$ = 59. C) N: $UAS\text{-}cyc^{DN}/+$ = 36, $cry[19]>cyc^{DN}$ = 44, $cry[19]\text{-}Gal4/+$ = 47. D-E) Box plots showing the slope of the evening peak on the last day of LD (D) and the 6th day of LLTC (E). The slope is calculated as follows: $(Act_{max}-Act_{ZTmin}) / (ZT_{max}-ZT_{min})$, with $ZT_{min}$ being the last time point of the minimum median value and $ZT_{max}$ the first time point with the maximum value before startle response to Zeitgeber change. N: $UAS\text{-}cyc^{DN}/+$ = 95 (B+C). For other genotypes see B-C. D) $UAS\text{-}cyc^{DN}/+$: $ZT_{min}$8.5, $cry[19]\text{-}Gal4/+$: $ZT_{min}$9.5, $Mai179>cyc^{DN}$, $Mai179\text{-}Gal4/+$,

$cry[19]{>}cyc^{DN}$: $ZT_{min}9$. E) UAS-$cyc^{DN}$/+ and $cry[19]{>}cyc^{DN}$: $ZT_{min}7$, $Mai179{>}cyc^{DN}$: $ZT_{min}7.5$, $Mai179$-Gal4/+: $ZT_{min}6.5$, $cry[19]$-Gal4/+: $ZT_{min}5.5$. UAS-$cyc^{DN}$/+: $ZT_{max}11.5$, $Mai179{>}cyc^{DN}$: $ZT_{max}11.5$ and $cry[19]{>}cyc^{DN}$: $ZT_{max}11$, $Mai179$-Gal4/+: $ZT_{max}10$, and $cry[19]$-Gal4/+: $ZT_{max}9$. Statistical test: Kruskal wallis [57]. *: $p{<}0.05$, **: $p{<}0.005$, ***:$p{<}0.001$. In the box plots, the lowest line indicates the first interquartile, the central line the median, the upper line the third interquartile, the cross the average, and the whiskers indicate the minimum and maximum, except for outliers.

observed normal PER oscillations, albeit with a slight amplitude decrease (Fig 2B). Normal PER cycling was also maintained in the other, largely CRY⁻ neurons, not expressing *Mai179* (Fig 2D), indicating that they receive independent temperature input. Flies with ablated morning and evening oscillator neurons also exhibit synchronized PER oscillations in the DN1-3 during temperature cycles in DD, further supporting the existence of multiple temperature inputs into the brain clock [37].

## EO neurons are necessary for synchronized behavior in constant light and temperature cycles

In order to restrict $cyc^{DN}$ expression exclusively to the EO we combined *Mai179-Gal4* with *Pdf-Gal80* (Materials and Methods). To test the efficiency of *Pdf-Gal80* repression, we dissected *Mai179-Gal4 Pdf-Gal80* flies expressing both GFP and $cyc^{DN}$ in LD at ZT2 and analyzed PER and GFP signals. The absence of GFP expression in the PDF⁺ cells confirmed the efficiency of *Pdf-Gal80*. Also, as expected, PER was absent from 2–3 CRY⁺ LNd, but in the 5th sLNv PER expression was similar as in the l-LNv (Fig 3A), suggesting that in this genotype the effect of $cyc^{DN}$ expression in the 5th s-LNv was not 100% penetrant. Behaviorally, in LLTC the amplitude of the evening peak was significantly reduced (Fig 3B–3D), demonstrating that a functional clock in the evening cells is necessary for synchronizing the evening activity peak in LLTC. Interestingly, although the slope is reduced in LD, *Mai179-Gal4 Pdf-Gal80* > $cyc^{DN}$ flies are still synchronized (S2A and S2C Fig), whereas in LLTC they are desynchronized (S2B and S2C Fig). The reduced, but otherwise synchronized slope in LD suggests a delayed activity peak, which is indeed visible on the first day in LL (Fig 3B). Next, we applied an even more restricted driver to disturb clock function in a subset of the EO neurons. *spE-Gal4* is expressed in the 5th s-LNv and the CRY⁺ LNd, but not in the DN1a [38]. Interestingly, expression of $cyc^{DN}$ does not alter the shape of behavioral evening activity in both LD and LLTC (S3 Fig). Although this may indicate a prominent role for the DN1a in regulating evening activity, we think the weak expression of the *spE-Gal4* driver allows for sufficient clock function in the 5th s-LNv and the CRY⁺ LNd evening cells of *spE* > $cyc^{DN}$ flies (see S5 Fig).

**Table 1. List of *Gal4* drivers and their expression pattern used in this study.**

| Line | Expression in | Reference |
|------|---------------|-----------|
| *Pdf-Gal4* | sLNv, lLNv | [51] |
| *DvPdf-Gal4* | sLNv, 5th sLNv, 3 LNd CRY⁻, 1LNd CRY⁺ | [49] |
| *Clk9M-Gal4* | sLNv, DN2 | [50] |
| *Mai179-Gal4* | sLNv, 5th sLNv, 3 LNd CRY⁺, DN1a | [11] |
| *Cry[19]-Gal4* | sLNv, 5th sLNv, 3 LNd CRY⁺, DN1a | [30] |
| *spE-Gal4* | 5th sLNv, 3 LNd CRY⁺ | [38] |
| *R16C05-Gal4* | DN1a, 2 LNd CRY⁺ | [39] |
| *Clk4.1M-Gal4* | DN1p | [18] |
| *spLPN-Gal4* | LPN neurons | [54] |
| *Clk856-Gal4* | all clock neurons | [52] |

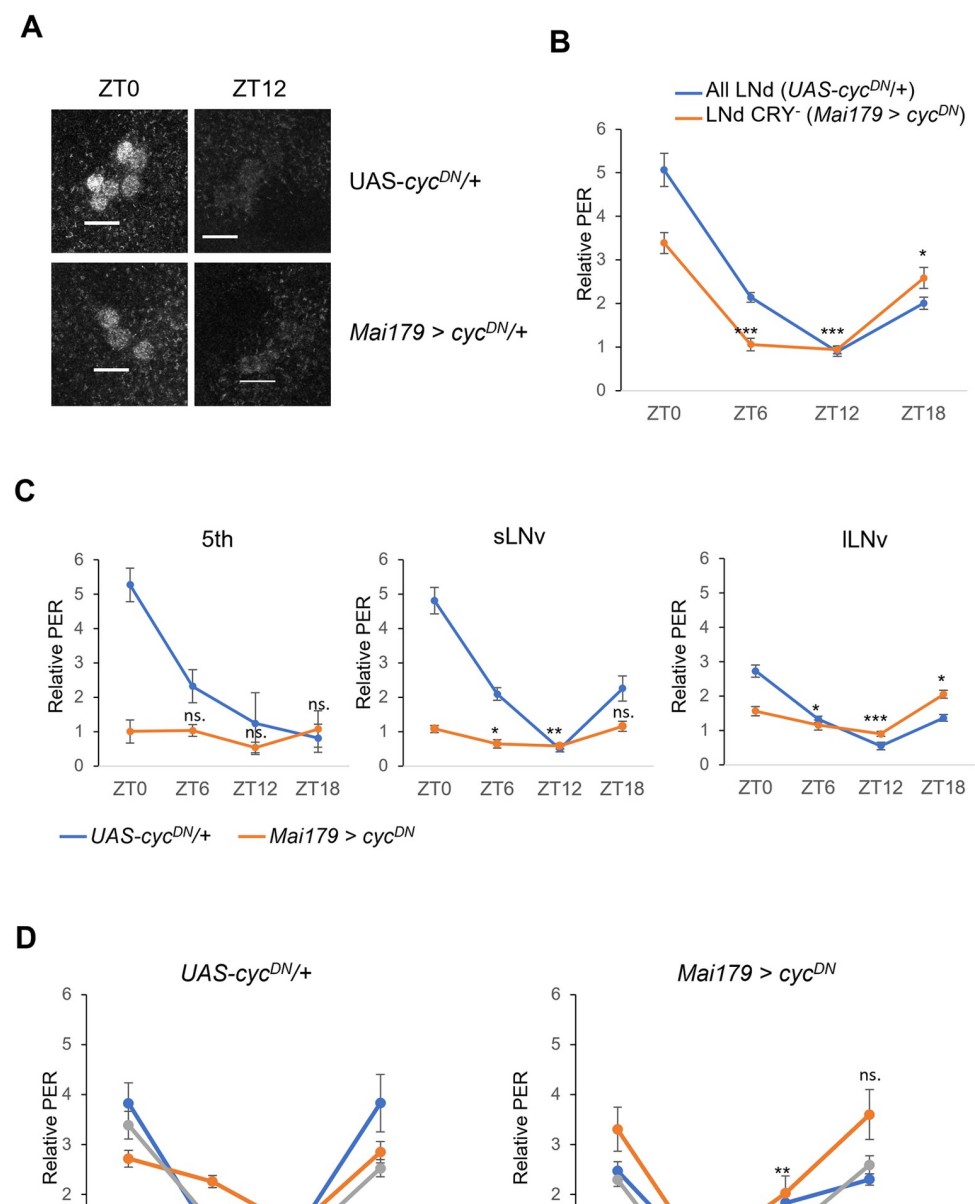

**Fig 2. Blocking clock function in *Mai179-Gal4* cells does not prevent temperature synchronization of PER expression in other clock neurons.** PER immunostaining on the 6th day of LLTC6. A) PER signals in the LNd at ZT0 and ZT12 in controls (*UAS-cyc^{DN}/+*) and *Mai179 >cyc^{DN}* brains. Scale bar: 10μm. B) Average relative PER levels in the LNd. Only the 3 LNd (presumably CRY⁻) were visible and quantified in *Mai179 >cyc^{DN}* brains, while all 6 LNd were quantified in the controls. C, D) Average relative PER levels in the other Lateral (C) and Dorsal Neurons (D).

To determine if clock function within the MO or other clock neurons is also required for the evening activity peak in LLTC, we screened additional clock neuronal *Gal4* drivers (S4B Fig and Table 1). Interestingly, only *Clk856-Gal4*, which drives expression of *cyc^{DN}* in all brain clock neurons, showed a significant decrease of the synchronized evening behavior, while

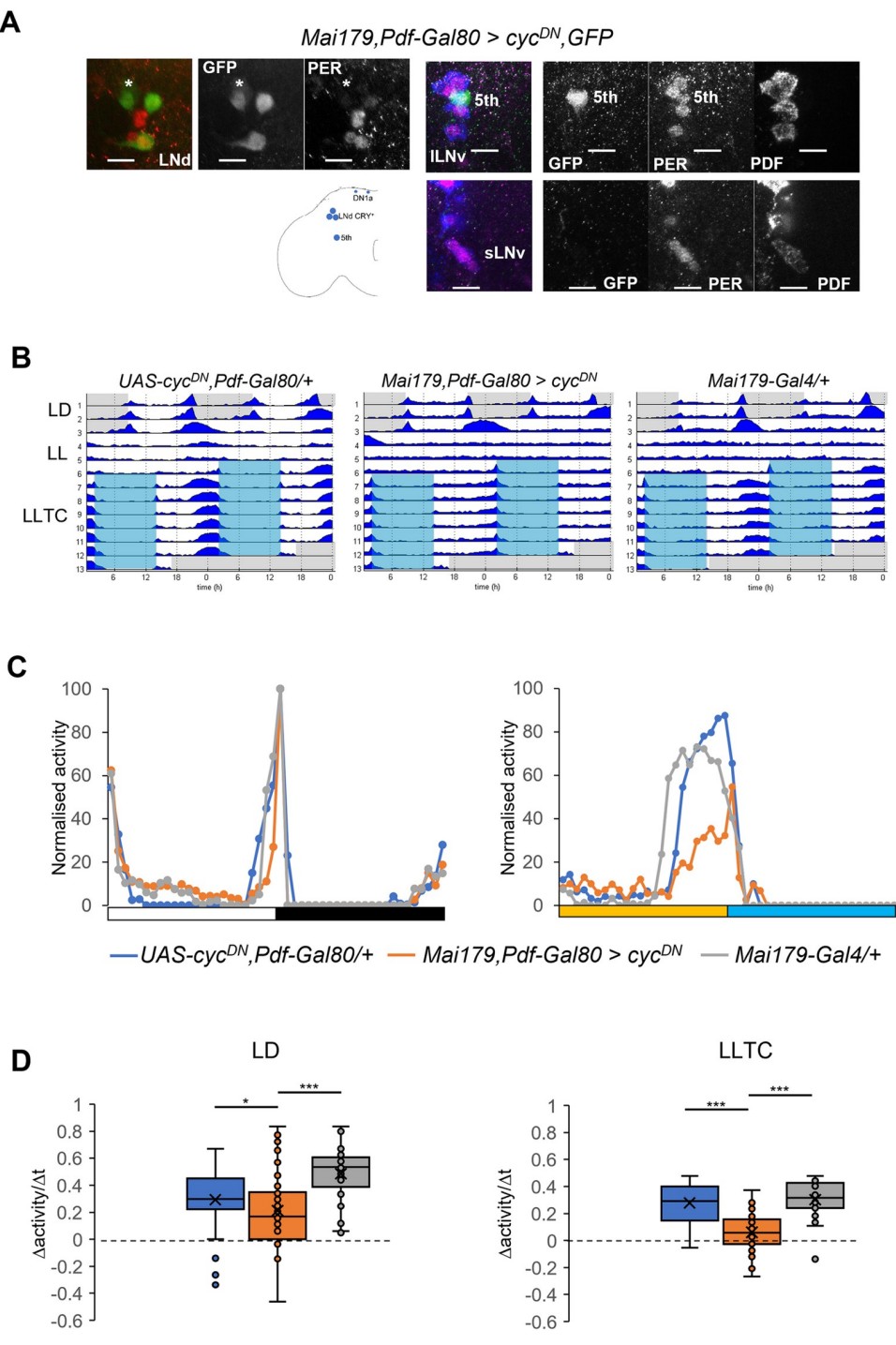

**Fig 3. Circadian clock disruption in EO neurons prevents synchronized behavior in temperature cycles during constant light.** A) Immunostaining of a *Mai179, Pdf-Gal80>cyc^{DN},GFP* brain at ZT2 in LD (25°C). GFP: green; PER: red; PDF: blue. Left panel: LNd. The * marks a non-LNd GFP+ cell. Scale bar: 10μm. Cartoon depicts clock neurons expressing *cyc^{DN}* in *Mai179, Pdf-Gal80* flies. B) Average actograms of one representative experiment. N: *UAS-cyc^{DN}*, *Pdf-Gal80/+* = 14, *Mai179, Pdf-Gal80>cyc^{DN}* = 31, *Mai179-Gal4/+* = 16. C) Median of the normalized locomotor activity during the last day of LD (left) and the 6th day of LLTC (right). N: *UAS-cyc^{DN},Pdf-Gal80/+* = 34, *Mai179, Pdf-Gal80>cyc^{DN}* = 48, *Mai179-Gal4/+* = 32. D) Box plots showing the slope of the evening peak on the last day of LD (left) and the 6th day of LLTC (right). Same flies as in (C). For all the genotypes in LD ZT_{min} = 9.5. In LLTC, *UAS-cyc^{DN},Pdf-Gal80/+* and *Mai179, Pdf-Gal80>cyc^{DN}*: ZT_{min}7.5 and *Mai179-Gal4/+*: ZT_{min}6. *UAS-cyc^{DN},Pdf-Gal80/+*: ZT_{max}11.5, *Mai179, Pdf-Gal80>cyc^{DN}*: ZT_{max}10.5, and *Mai179-GAL4/+*: ZT_{max}9.5.

none of the other drivers showed significant differences compared to the respective controls in LLTC. In particular, neither *Pdf-Gal4* nor *DvPdf-Gal4* which are both expressed in all of the PDF[+] MO neurons, had an effect on the evening peak (S4 Fig). In agreement with previous findings [22], this suggests that a functional clock in the MO is not necessary for synchronizing behavior in LLTC. Moreover the *DvPdf-Gal4* result shows that expression of *cyc[DN]* in only two out of the six EO neurons (1 CRY[+] LNd and the 5[th] s-LNv, Table 1) is not sufficient to interfere with synchronizing evening activity in LLTC (S4 Fig).

## Cell ablations confirm the role of evening neurons for synchronization to temperature cycles in constant light

To underpin the role of the EO in controlling evening activity, we ablated these neurons using the Gal4/UAS system to drive expression of the pro apoptotic genes *head involution defective* (*hid*) and *reaper* (*rpr*). Because of the broad expression pattern of *Mai179-Gal4* outside the brain clock, activation of *UAS-hid,rpr* using this driver led to lethality. In contrast, *cry[19]>hid,rpr* flies are viable and show the almost complete disappearance of the evening peak in LD and in LLTC (Figs 4 and S5A–S5C). Because *cry[19]-Gal4* is also expressed in the morning oscillator neurons, we next used *spE-Gal4* in order to ablate the CRY[+] LNd and the 5[th] s-LNv [38, 39]. Similar to the *spE>cyc[DN]* experiments, there was no obvious difference between *spE>hid,rpr* flies and their parental controls in LD and the slope was not significantly different (Figs 4A, S5A and S5B). However, compared to controls, *spE>hid,rpr* flies showed a reduced Slo$_{exp}$/Slo$_{the}$ ratio (0.67 for *spE>hid,rpr* compared to 0.86 and 0.89 for both parental controls, respectively), which is correlated with a 15% increase of flies showing non-anticipatory behavior (S5C Fig). During LLTC, *spE>hid,rpr* flies show a strongly reduced amplitude of the evening activity, accompanied by an increase of the percentage of desynchronized flies (Fig 4A–4D), confirming the role for the evening neurons during temperature synchronization in constant light. To determine ablation efficiency we dissected *spE>hid,rpr* flies in LD at ZT2-ZT3 and immunostained the brains with PER as a marker. Although not 100% penetrant, in the majority of brains only 3 to 4 LNd could be detected, presumably representing the CRY[-] LNd subset. In addition, the 5[th] s-LNv was detectable in 50% of the hemispheres, although it was never detectable in both hemispheres of the same brain (S5D and S5E Fig). Hence, although the incomplete penetrance of this driver is most likely responsible for the weak behavioral phenotype in LD, the stronger phenotype observed in LLTC indicates a more prominent role for the CRY[+] PDF[-] neurons in LLTC compared to LD.

To confirm that the MO neurons are not required for the synchronization to LLTC, we induced their ablation by expressing *hid* and *rpr* using *Pdf-Gal4* and *DvPdf-Gal4*. As a positive control, we also ablated all clock neurons using *Clk856-Gal4*. Indeed, *Clk856 > hid,rpr* were the only flies that were completely desynchronized both in LD and LLTC (S6A and S6C Fig). As expected, both *Pdf >* and *DvPdf > hid,rpr* flies revealed altered behavior in LD, revealing the typical lack of morning anticipation and advance of the evening peak (S6A and S6B Fig). Interestingly, we also observed increased desynchronization in LD, particularly with the *Pdf-gal4* driver (S6C Fig). During LLTC, both *Pdf >* and *DvPdf > hid,rpr* flies showed robust synchronization (S6 Fig). The slightly decreased synchronization after *DvPdf-Gal4* ablation can be explained by the expression of this driver in a small subset of EO neurons (5[th] s-LNv and 1 CRY[+] LNd). In summary, the ablation experiments strongly support an essential role for the EO neurons for synchronizing fly behavior to temperature cycles in LL.

## Role of EO neurons in temperature cycles during constant darkness

Flies behave differently in LLTC compared to DDTC, even though the temperature regime remains the same [29]. For example in DDTC (25˚C:16˚C), flies are synchronized to the

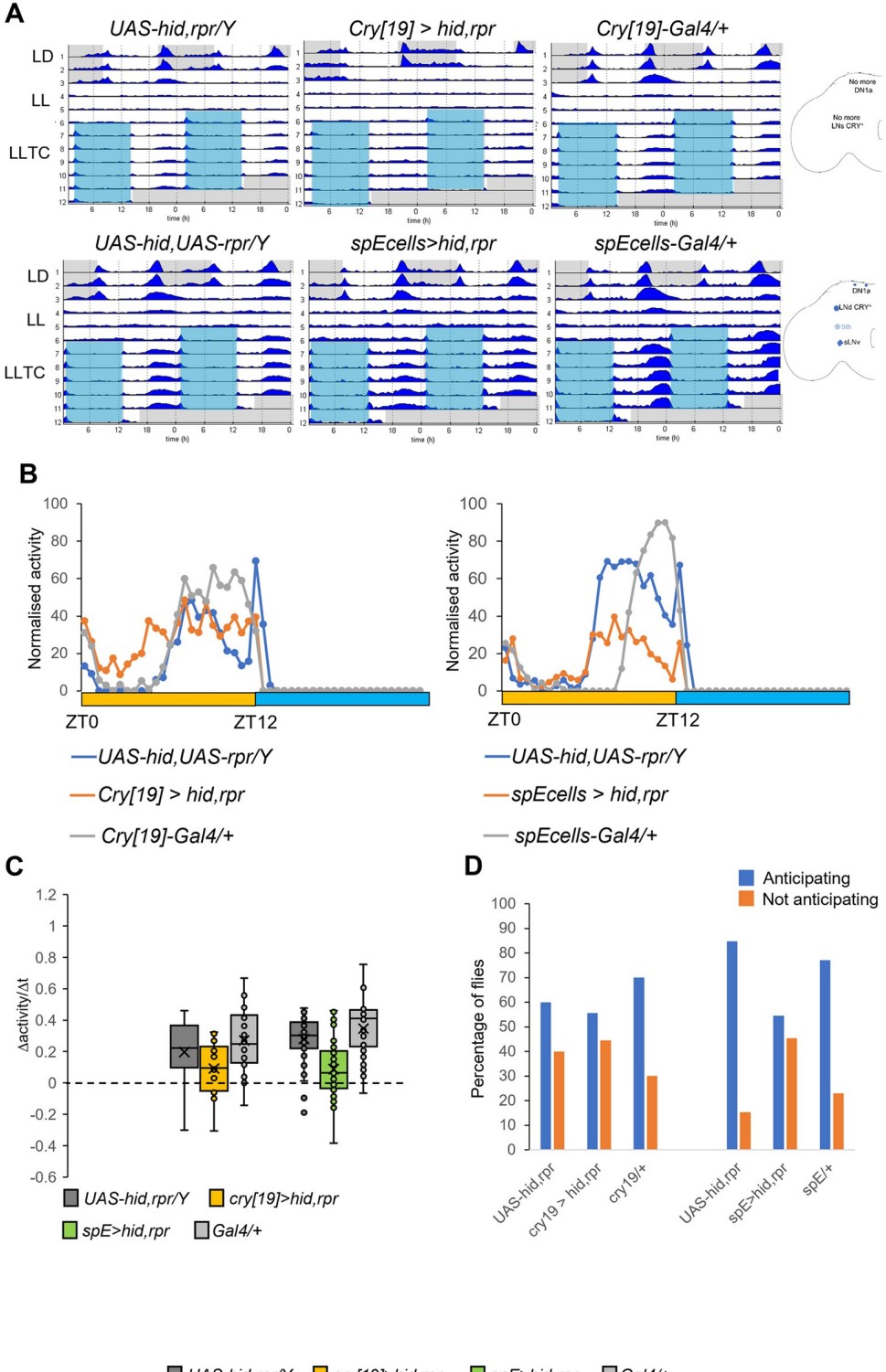

**Fig 4. EO neurons are required for behavioral synchronization to temperature cycles in constant light.** A) Double plotted average actograms of the indicated genotypes. N: *UAS-hid,rpr/Y; ls-tim)* and *cry-Gal4[19]/+* = 20, *cry[19]>hid, rpr* = 18, *UAS-hid,rpr/Y;s-tim/ls-tim* = 20, *spE-GAL4/+* = 19, *spE > hid,rpr* = 19. Cartoons indicate clock neurons that remain after *hid,rpr*-induced ablation. B) Median of normalized locomotor activity during day six of LLTC. Left panel: same flies as in A. N for right panel: *UAS-hid,rpr/Y, s-tim/ls-tim* = 59, *spE>hid,rpr* = 55; *spE-Gal4/+* = 61. C) Box plots showing the slope of the evening peak on the 6th day of LLTC. Same flies as in B. C) ZT$_{min}$ for all the genotypes is

ZT9.5, except for *spE>hid,rpr* ($ZT_{min}$ = 10). For left group *UAS-hid,rpr/+* and *cry-Gal4[19]/+*: $ZT_{min}$4.5, and $ZT_{min}$2.5 for *cry[19]>hid,rpr*. For right group *UAS-hid,rpr/+* and *spE>hid,rpr*: $ZT_{min}$5, and *spE-Gal4/+*: $ZT_{min}$7.5. For left group *UAS-hid,rpr/+*: $ZT_{max}$7.5, and *cry[19]>hid,rpr* and *cry[19]-GAL4/+*: $ZT_{max}$7. For right group ZT8.5 for *UAS-hid,rpr/+* and *spE>hid,rpr*: $ZT_{max}$8.5, and *spE-Gal4/+*: $ZT_{max}$11. D) Percentage of flies anticipating (blue) and not anticipating (orange) lights-off in LD (left) or the temperature decrease (right). Same flies as in B.

beginning, while in LLTC they are synchronized to the end of the thermophase ([29], Fig 1). If, depending on the light condition, the neuronal network balance switches from one state to another, the EO should not be operating during DDTC. To test this, we interfered with clock function by expressing *cyc^DN* in both the MO and EO neurons (*cry[19]-Gal4 > cyc^DN*) or only in the EO (*cry[19]-Gal4, Pdf-Gal80 > cyc^DN*). Strikingly, while *cry[19]Gal4 > cyc^DN* flies exhibit a very broad activity peak, covering almost the entire warm phase, flies with *cyc^DN* expression restricted to the EO have their activity peak in the 1st half of the warm phase, indistinguishable from the controls (Fig 5A and 5B), and suggesting that the MO controls behavior under these conditions [29, 37]. In addition, while in DD and constant temperature *cry[19]-Gal4 > cyc^DN* flies become arrhythmic, due to the expression of *cyc^DN* in the s-LNv pacemaker neurons [31], *cry[19]-Gal4, Pdf-Gal80 > cyc^DN* exhibit synchronized rhythmic activity (Fig 5A and 5C). This demonstrates both, the effectivity of *Pdf-Gal80*-mediated repression of *cyc^DN* expression in the s-LNv morning cells, and the previous synchronization to temperature cycles (based on the maintenance of the activity phase obtained during DDTC). Because *Mai179-Gal4* expression is weaker in the MO compared to the EO neurons [11], we also analyzed *Mai179>cyc^DN* in DDTC: Strikingly, while synchronization was strongly reduced in LLTC (Fig 1), *Mai179>cyc^DN* flies exhibited robustly synchronized activity rhythms in DDTC (S7 Fig), however with a phase delay compared to controls (S7A and S7B Fig). Nonetheless, after switching to DD, flies free run with the same phase as during the previous DDTC regime, indicating stable synchronization of the circadian clock and confirming weak s-LNv expression of this driver (S7A and S7C Fig). Restricting *Mai179 > cyc^DN* expression to the EO, by introducing *Pdf-Gal80*, restored the phase to the beginning of the thermophase as in controls (S7A and S7B Fig), showing the weak *cyc^DN* expression in the s-LNv is enough to change the phase in DDTC. In summary, these results show that while the EO controls behavioral synchronization during LLTC, the MO takes over this role in the absence of light.

## Discussion

The daily pattern of locomotor behavior is highly plastic and not only depends on the time of day but also on the current environmental condition. How the brain clock integrates Zeitgeber information and various environmental inputs to time locomotor activity is not clear. To address this important question, we used a specific daily temperature oscillation (25˚C-16˚C) as Zeitgeber and light input as the environmental variable. By genetically probing clock- and neuronal function of different parts of the circadian neuronal network in different environmental conditions, we reveal that the ambient light status determines circadian network balance.

### Phase control by the environment

An important question in chronobiology that so far remains unanswered is the mechanism of entrainment by light and temperature when these two environmental parameters are actually not so reliable considering their substantial daily variation [40]. Notably, substantial weather-inflicted variations of temperature and light intensity can occur during the day and these must not lead to circadian clock resetting. Nonetheless, animals behave quite differently depending on the current environmental status regardless of their clock entrainment status. For example,

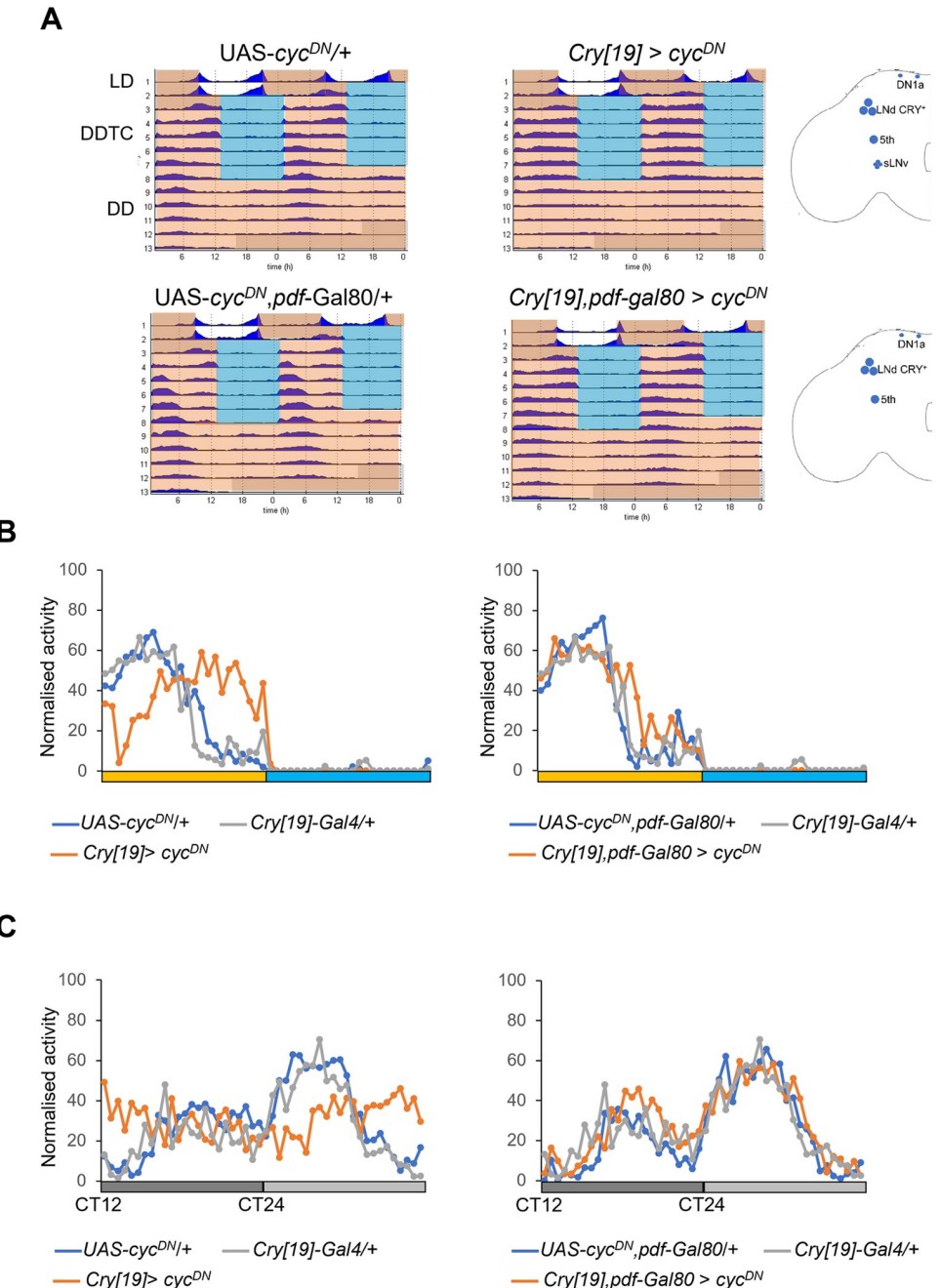

**Fig 5. Clock function in the MO and EO neurons is not required for synchronization to temperature cycles in constant darkness (DDTC).** A) Double plotted average actograms of the indicated genotypes. N: *UAS-cyc*$^{DN}$/+ = 25, *cry[19]>cyc*$^{DN}$ = 30; *UAS-cyc*$^{DN}$,*pdf-Gal80/+* = 21; *cry[19]*,*pdf-Gal80 > cyc*$^{DN}$ = 25. Conditions are indicated to the left. White areas indicate lights on and 25˚C, orange areas lights off and 25˚C, blue area light off and 16˚C. Cartoons to the right indicate clock neurons expressing *cyc*$^{DN}$ in the respective genotypes. B-C) Median of normalized locomotor activity during the 6$^{th}$ day of DDTC (B), and the first two DD days in constant conditions (C). Same flies as in A. The dark grey bars represent the subjective night of DD1, the light grey bars subjective day of DD2.

fruit flies lose their two anticipatory activities present in LD 25˚C at different constant temperatures. At warm temperature (≥30˚C), flies only anticipate the light-on transition while at colder temperatures (<20˚C) they only anticipate the lights-off [41]. On the other hand, when

temperature cycles are used as Zeitgeber in constant darkness, the absolute temperatures determine the phase of the activity peak, with TCs of 25°C:16°C and 29°C:20°C inducing activity in the first half or the second half of the thermophase, respectively [29, 37]. In constant light however, both temperature cycles result in an activity peak during the second half of the thermophase [29].

Here, we demonstrate that the EO that drives the evening peak in LD25°C is necessary to drive the evening peak in LLTC25°C-16°C, while the MO that drives the morning peak in LD25°C controls the phase in DDTC25°C-16°C. According to our model the ambient environmental condition modulates the clock network balance (Fig 6). Hence, the environmental condition must be taken into account in order to understand the role of each oscillator. Previously, trying to understand how temperature cycles synchronize flies, a similar approach has been performed, although in the absence of light and different absolute temperatures, but with the same amplitude of temperature oscillations (29°C-20°C). Consistent with our findings, ablating both the MO and EO using a *cry-gal4* line lead to largely desynchronized behavior [37] (Fig 5). As mentioned above, in DDTC 29°C:20°C conditions flies exhibit an afternoon activity peak, and ablating the PDF neurons does not affect this synchronization [37]. In contrast, our data show that a functional clock in the PDF neurons is important for synchronized morning activity in DDTC 25°C:16°C (Figs 5 and S7). Following our model, we postulate therefore that the DN1p clock neurons also contribute to the behavioral evening activity during warm temperature cycles in constant darkness. The DN1p can drive evening activity during DDTC 29°C:20°C and during low intensity LD cycles when the temperature is constant, while at high light intensity LD cycles they support morning activity [18]. Furthermore, the DN1p receive warm input via the TrpA1 expressing AC neurons [42]. Interestingly however, TrpA1 is not required for temperature entrainment as such, but shapes behavior under warm conditions [42–44]. Notably, while the loss of TrpA1 function has no effect on the behavioral phase in DDTC 25°C:16°C, TrpA1 mutant flies show an advanced increase of activity and a reduced siesta in DDTC 29°C:20°C compared to controls [43]. Hence, taken together these data re-enforce our model where, depending on the environmental condition, the clock network can change its balance to set behavioral activity phase, bypassing a change of clock entrainment (Fig 6).

Furthermore, our data suggest that the length of the light period (here 12 h vs 24 h) influences the involvement of the EO. Ablating the PDF neurons not only advances the evening peak in LD but it also drastically reduces the synchronization of the flies (S6B and S6C Fig), while they remain strongly synchronized in LLTC. Furthermore, while ablating the EO, even with incomplete penetrance, drastically affected the presence of the synchronized evening peak in LLTC, it only slightly reduced synchronization in LD. Hence, although the clock of the EO can be entrained independently to light [11], our data indicate that the PDF neurons have a strong influence on the robustness of EO-output in LD25°C but not in LLTC. This suggests that the extent of dark periods increases the influence of the PDF neurons during the day. The evening peak occurs under the same environmental condition (lights-on and 25°C), however with a slight advance in LLTC compared to LD (Fig 1). Therefore, we propose that the past experience during the night/cryophase determines the influence of one group toward the other. In summary, these data confirm that the network status varies with the environmental condition, and therefore it is crucial to consider the specific environmental conditions when deciphering the different contributions of circadian network components in regulating behavioural activity. Our simple environmental protocol provides a template to further test and extend this model. For example, using temperature cycles as Zeitgeber, we can now modify light quality and intensity to test how the clock network responds to these light variations while the clock is stably entrained.

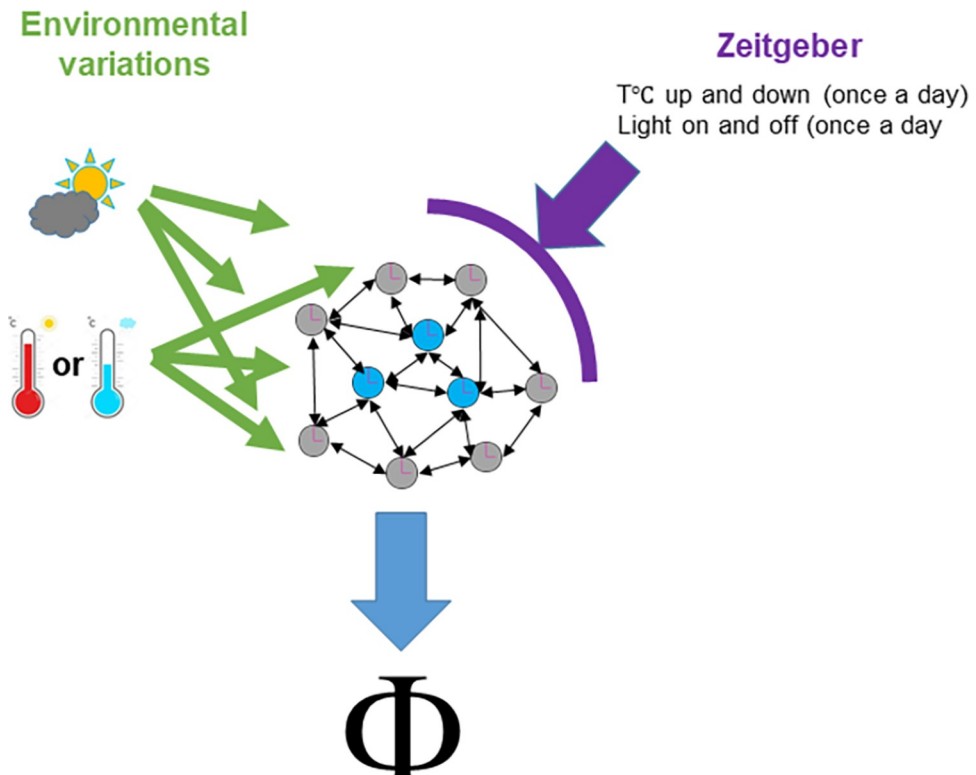

**Fig 6. Environmental cues act as Zeitgeber and set behavioral activity Phase.** The molecular clock in brain clock neurons is entrained by light and temperature. However, these two environmental inputs vary on a day to day basis (e.g., cloudy versus sunny days), and animals change their activity at different times of day in response to the ambient environmental condition. How does the circadian system distinguish between an input that entrains the circadian clock (Zeitgeber) and one that after integration by the clock system, sets the daily activity phase? Here, we demonstrate that the EO determines the behavioral evening peak (Φ) in the presence of light and 25˚C, while the circadian clock is entrained with a 25˚C:16˚C temperature cycle. In contrast, the MO determines the morning peak in constant darkness and 25˚C, during the same temperature cycle. We present a model explaining how the environment and the circadian clock shape locomotor activity. On one hand, the on/up and off/down environmental changes that happen once a day entrain the molecular clocks in the system (purple arrow). On the other hand, different environmental input such as light (quality and intensity) and temperature (different levels), here represented by green arrows, are perceived by different oscillators in different manners. Depending on the time of day (the molecular clock status of the system), this will lead to a modification of the network balance and a dominancy of one or several oscillators (blue clocks *vs* grey ones), resulting in a behavioral activity phase (Φ) according to the internal timing and the current condition. Therefore, to biologically demonstrate this model, we fixed a Zeitgeber (in this study TC 25˚C-16˚C) and tested how light (here presence/absence) modifies the balance. From this basis, we can now apply more subtle modifications such as the intensity or the quality of light to change the phase of the behavior and use this to understand the principles underlying neuronal network switches.

## Circadian clock entrainment by temperature cycles

Synchronization of clock protein oscillations in LPN neurons is preferentially sensitive to temperature [21–23]. However, we have previously observed that these neurons are not necessary for rhythmic behavior under TC conditions in both LL and constant darkness [29]. Here, we confirm the non-requirement of clock function within the LPN for rhythmic locomotor behavior in LLTC (S4B Fig), suggesting that LPN sensitivity for molecular synchronization to TC serves another function unrelated to entrainment. Indeed, a recent study shows that the LPN are activated above 27˚ and they are important for increased siesta sleep at 30˚C [45]. CRY is expressed in about 1/3rd of the clock network in the brain and plays an important role in light entrainment [7, 46], consistent with the idea that CRY+ neurons are more sensitive to

light and CRY⁻ cells are more sensitive to temperature [23]. Notably, isolated brains can still be entrained to LD cycles [47], and in the absence of visual input all clock neurons are entrained to LD [46], indicating that the CRY⁺ neurons synchronize the remaining CRY⁻ neurons non-cell autonomously [48]. However, here we demonstrate an essential role of the CRY⁺ neurons in controlling rhythmic behavior in LLTC. Nevertheless, even in the absence of all CRY⁺ neurons, flies exhibit weak synchronized behavior with an evening phase during temperature cycles (in LL and DD), suggesting that CRY⁻ clock neurons (subsets of DN1$_p$ and LNd, DN2, and DN3) also play a role in temperature synchronization [29, 37]. Nevertheless, because they are not able to instruct the CRY⁺ neurons during TC (Fig 2), their role in temperature entrainment is not as prominent as that of CRY⁺ neurons in LD (see above). Moreover, ablation of PDF⁺ or all three groups of LN (LNv, LNd, LPN) still allows for weak behavioral evening synchronization to 30˚C: 25˚C TC in LL [22]. Furthermore, the fact that in the absence of a clock in the Mai179⁺ cells (Fig 2), after ablation of all CRY⁺ cells [37], or in the absence of all LN [22], molecular oscillations in other clock neurons can be synchronized, supports the idea that multiple and independent pathways contribute to temperature entrainment [32, 34–36].

## Material and methods

### Fly strains

Flies were raised in a 12 h:12 h light dark (LD) cycle on a medium containing0.7% agar, 1.0% soy flour, 8.0% polenta/maize, 1.8% yeast, 8.0% malt extract, 4.0% molasses, 0.8% propionic acid, and 2.3% nipagin at 25˚C and 60% relative humidity. The following fly lines were used in this study: *Mai179-GAL4/CyO* [11], *cry[19]-GAL4* [30], *spE-Gal4* (JRC-MB122B) [38], *DvPdf-GAL4* [49], *R16C05-Gal4* (BL69492) [39], *Clk4.1M-GAL4* [17, 18], *Clk9M-Gal4* [50], *Pdf-GAL4* [51], *Clk856-GAL4* [52];*UAS-cyc^DN* [31], *Pdf-GAL80* (BL80940), GFP-*cry* [53]. *spLPN-GAL4* (*R11B03-p65.AD; R65D05-GAL4.DBD*) [54] was provided by Taishi Yoshii. *Mai179-GAL4*, *UAS-cyc^DN*, *Clk9M-GAL4*, and *Clk4.1-Gal4* have been outcrossed for 5 generations to *iso31* to standardize the genomic background [55]. We recently showed that only flies carrying at least one copy of the natural *ls-tim* allele, encoding both the light sensitive S-TIM and less light sensitive L-TIM protein [27] are able to synchronize to temperature cycles in constant light, while flies homozygous for the *s-tim* allele (encoding only S-TIM) cannot [27]. Therefore, all strains analyzed in this study carry at least one copy of *ls-tim*.

### Behavior

Analysis of locomotor activity of male flies was performed using the Drosophila Activity Monitoring System (DAM2, Trikinetics Inc., Waltham, MA, USA) with individual flies in recording tubes containing food (2% agar, 4% sucrose). Briefly, DAM2 activity monitors containing LD-entrained flies were placed inside a light- and temperature-controlled incubator (Percival Scientific Inc., Perry, IA, USA). Fly activity was monitored for at least 11 days with the first 2 days in LD25˚C (700–1000 lux generated by 17W F17T8/TL841 cool white Hg compact fluorescent lamps, Philips) followed by three days in LL 25˚C and then LLTC 25˚C-16˚C with a shift delayed by 5h relative to the previous LD. The LD behavior analysis was performed on the last day of LD, while LLTC behavior was analyzed on the 6^th day of LLTC.

Plotting of actograms was performed using a signal-processing toolbox [56] implemented in MATLAB (MathWorks, Natick, MA, USA). The 24h locomotor activity plots were generated using a custom excel macro [42]. Activity was averaged into 30 minute bins and normalized to the maximum individual activity level. The median of this normalized activity was plotted, because it is a more representative parameter of behavioral synchrony compared to the mean. To measure the slope, we manually determined the latest time point of the

minimum median ($ZT_{min}$) and the maximum median ($ZT_{max}$) before Zeitgeber transition. The only exception is shown in S7C Fig, where the downhill slope was analyzed in the subjective day of DD2, because controls show a better synchronization while decreasing their activity, compared to the period when activity increases. We calculated for each fly the derivative of the line between $ZT_{max}$ and $ZT_{min}$: Slope = ($Act_{max}$- $Act_{ZTmin}$)/($t_{ZTmax}$-$t_{ZTmin}$). Time was counted in minutes, but rounded to full hours in figures and legends for convenience. The box plots were made using Excel. The statistical tests were performed using the freely available Estimation Statistics [57].

## Immunostaining

Adult male *Drosophila* brains were immuno-stained as described previously [55]. Briefly, brains were fixed in 4% paraformaldehyde for 20 min at RT, and blocked in 5% goat serum for 1 h at RT. Primary antibodies used were as follows: rabbit anti-DsRed (Clontech)– 1:2000; mouse anti-PDF (Developmental Studies Hybridoma Bank, DSHB)– 1:2000; mouse anti-Bruchpilot (nc82, DSHB)– 1:200; chicken anti-GFP (Invitrogen)– 1:1000 –rabbit anti-PER [58] 1:15000. Alexa-fluor secondary antibodies (goat anti-rabbit 555, goat anti-chicken 488, goat anti-mouse 488; Invitrogen) were used at 1:2000 except for labeling anti-BRP where goat anti-mouse 647 where a dilution of 1:500 was used. Confocal images were taken using an inverted Leica LSP8. PER signal intensity was quantified using an 40x oil-objective and imageJ. Pixel intensities were normalized to the background after background subtraction [(signal-background)/background] using Excel.

## Supporting information

**S1 Fig. MO and EO circadian clocks are required for synchronization to temperature cycles in constant light.** A) Representative median of the normalized activity of control flies in LD. The white rectangle delimits the 12h of light while the grey box delimits the12h of darkness. This example explains the calculation of the slope (theoretical, $Slo_{the}$, and experimental, $Slo_{exp}$). $Slo_{the}$ is calculated from the median levels according to the formula ($Act_{ZTmax}$- $Act_{ZTmin}$)/($t_{ZTmax}$-$t_{ZTmin}$) explained in Materials and Methods. For example here, the median activity of the first maximum before light-off (ZT11.5) is 67.6. Hence, here $Slo_{the}$ = (67.6–0)/(691–511) = 0.38. The calculation is made with the ZT in minutes. $Slo_{exp}$ is the median of the individual calculated slopes. B) Values of the $Slo_{the}$, $Slo_{exp}$ and the ratio $Slo_{exp}$/$Slo_{the}$ of the indicated genotypes in LD and LLTC6. If fly activity is highly synchronized between individuals, the ratio of $Slo_{exp}$/$Slo_{the}$ is close to 1 (i.e., most individuals behave similar to the median), while for desynchronized populations the $Slo_{exp}$/$Slo_{the}$ ratio is smaller (i.e., many individuals deviate from the median). C) In addition, we calculated the percentage of flies for each genotype with a $Slo_{exp}$ > ½ of the $Slo_{the}$ where a high versus low percentage again indicates synchronized or desynchronized behavior, respectively. Indeed, > 80% of the control flies increase their locomotion with $Slo_{exp}$ > ½ $Slo_{the}$, compared to only 50–60% of the *Mai179>cyc^{DN}* and *cry[19]>cyc^{DN}* flies (S1C Fig, blue bars percentage of flies with $Slo_{exp}$ > ½ of $Slo_{the}$ 'anticipating', orange bars percentage of flies with $Slo_{exp}$ < ½ of $Slo_{the}$ 'non-anticipating'). Same flies as in Fig 1B–1E).
(TIF)

**S2 Fig. Circadian clock disruption in EO neurons prevents synchronized behavior in temperature cycles during constant light.** A-B) $Slo_{the}$, $Slo_{exp}$ values and the ratio $Slo_{exp}$/$Slo_{the}$ of the indicated genotypes in LD (A) and on day six of LLTC (B). C) Percentage of flies anticipating (blue) and not anticipating (orange) lights-off (left) or the temperature decrease (right),

defined as described in the legend of S1 Fig and in Materials and Methods. Same flies as in Fig 3C–3D.
(TIF)

**S3 Fig. Stopping the clock with the *spE-Gal4* driver does not prevent synchronization to light:dark or temperature cycles.** A) Double plotted average actograms of the indicated genotypes. Cartoon on the right shows the clock neurons expressing *spE-Gal4*. N: *UAS-cyc$^{DN}$/+* = 19, *spE > cyc$^{DN}$* = 20, *spE-Gal-4/+* = 19. B) Median of normalized locomotor activity (left) and slope (right) in LD and LLTC6. N: *UAS-cyc$^{DN}$/+* = 58, *spE > cyc$^{DN}$* = 60, *spE-Gal-4/+* = 46. Statistical test: Kruskal wallis [57]. $^*$: p<0.05, $^{**}$:p<0.005, $^{***}$:p<0.001. C) Values of the Slo$_{the}$, Slo$_{exp}$ and the ratio Slo$_{exp}$/Slo$_{the}$ of the indicated genotypes in LD and LLTC6. Same flies as in B. D) Percentage of flies anticipating (blue) and not anticipating (orange) lights-off (left) or the temperature decrease (right), defined as described in the legend of S1 Fig and in Materials and Methods. Same flies as in B.
(TIF)

**S4 Fig. Stopping the clock in non-EO neurons does not affect the circadian activity in LLTC.** A) Double plotted average actograms of the indicated genotypes. Cartoon on the right shows the clock neurons expressing *DvPdf-Gal4*. Note the LNd CRY$^-$ are in red. N: *UAS-cyc$^{DN}$/+* = 18, *DvPdf > cyc$^{DN}$* = 20, *DvPdf-Gal-4/+* = 19. B) Box plots showing the slope of the evening peak on the 6$^{th}$ day of LLTC. N: *UAS-cyc$^{DN}$/+* = 172, 20 ≤ *Gal4>cyc$^{DN}$* ≤ 41, 18 ≤ *Gal4/+* ≤ 40.
(TIF)

**S5 Fig. EO neurons are required for behavioral synchronization to temperature cycles in constant light.** A) Median of normalized locomotor activity in LD. Same flies as in Fig 4B. B) Box plots showing the slope of the evening peak in LD. Same flies as in Fig 4B and 4C) Percentage of flies anticipating (blue) and not anticipating (orange) lights-off in LD. D) Immunostaining of lateral neurons. *spE-Gal4* drives expression of the nuclear marker dsRed, CRY$^+$ neurons are labeled by expression of a GFP-CRY fusion protein [53]. Scale bar 10μm E) Immunostaining and quantification to determine the number of ablated neurons after *hid* and *rpr* expression in spE cells. Flies were dissected at ZT2 in LD. The graph on the right shows the average number of LNd and 5$^{th}$-sLNv cells observed. Number of brains per genotype is 11.
(TIF)

**S6 Fig. MO neurons are not required for synchronization to temperature cycles in constant light.** A) Double plotted average actograms of the indicated genotypes. Cartoon on the right showing morning and evening clock neurons, remaining after ablation using the *DvPdf* driver. N: *UAS-hid,rpr/Y;ls-tim* = 21, *Clk856>hid,rpr* = 24, *DvPdf>hid,rpr* = 24. B) Median of normalized locomotor activity in LD (left) and during day six of LLTC (right). N: *UAS-hid,rpr/Y;ls-tim* = 34, *Pdf > hid,rpr* = 50, *Pdf-Gal4/+* = 34, *DvPdf>hid,rpr* = 28, *DvPdf-Gal4/+* = 44. C) Percentage of flies anticipating (blue) and not anticipating (orange) lights-off in LD (left) or the temperature decrease (right). Same flies as in B, and N for *Clk856>hid,rpr* = 24, *clk856-Gal4/+* = 44.
(TIF)

**S7 Fig. Clock function in EO neurons is not required for synchronization to temperature cycles in constant darkness.** Double plotted average actograms of the indicated genotypes. N: *UAS-cyc$^{DN}$/+* = 20, *Mai179>cyc$^{DN}$* = 19, *Clk856>cyc$^{DN}$* = 31, *Mai179,pdf-Gal80>cyc$^{DN}$* = 14. B) Median of normalized locomotor activity during the 6$^{th}$ day of DDTC. N: *UAS-cyc$^{DN}$/+* = 20, *Mai179>cyc$^{DN}$* = 19, *Mai179-Gal4/+* (left) = 20, *UAS-cyc$^{DN}$,pdf-Gal80/+* = 21, *Mai179,pdf-*

*Gal80>cyc$^{DN}$* = 14, *Mai179-Gal4/+* (right) = 17. C) Percentage of flies decreasing their locomotion with a slope steeper (blue), or lower (orange) than 50% of the theoretical slope after the morning peak at DD2 following DDTC. Same flies as in panel B and Fig 5C.
(TIF)

**S1 Data. Data sheet with all data used to graph main and supplemental Figures.** Excel file containing all data used for generating main and supplemental Figure graphs as well as statistical tests.
(XLSX)

## Acknowledgments

We thank Paul Hardin, François Rouyer, and Taishi Yoshii for providing fly strains.

## Author Contributions

**Conceptualization:** Ralf Stanewsky, Angelique Lamaze.

**Data curation:** Clara Lorber, Solene Leleux, Angelique Lamaze.

**Formal analysis:** Clara Lorber, Solene Leleux, Angelique Lamaze.

**Funding acquisition:** Ralf Stanewsky, Angelique Lamaze.

**Investigation:** Clara Lorber, Solene Leleux, Angelique Lamaze.

**Methodology:** Angelique Lamaze.

**Project administration:** Angelique Lamaze.

**Supervision:** Ralf Stanewsky, Angelique Lamaze.

**Validation:** Angelique Lamaze.

**Visualization:** Angelique Lamaze.

**Writing – original draft:** Ralf Stanewsky.

**Writing – review & editing:** Ralf Stanewsky, Angelique Lamaze.

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
