## [Decision Letter · Decision Letter 0]

9 Jul 2022

Dear Dr Stanewsky,

Thank you very much for submitting your Research Article entitled 'Light-dependent network switching between circadian morning and evening oscillators controls behavior during daily temperature cycles' to PLOS Genetics.

The manuscript was fully evaluated at the editorial level and by independent peer reviewers. The reviewers appreciated the attention to an important problem, but Reviewer 2 has raised some valid concerns about the current manuscript. Based on the reviews, we will not be able to accept this version of the manuscript, but we would be willing to review a much-revised version. We cannot, of course, promise publication at that time.

If you decide to revise the manuscript for further consideration at PLOS Genetics, please aim to resubmit within the next 60 days, unless it will take extra time to address the concerns of the reviewers, in which case we would appreciate an expected resubmission date by email to plosgenetics@plos.org.

[LINK]

We are sorry that we cannot be more positive about your manuscript at this stage. Please do not hesitate to contact us if you have any concerns or questions.

Yours sincerely,

John Ewer

Associate Editor

PLOS Genetics

Gregory P. Copenhaver

Editor-in-Chief

PLOS Genetics

Reviewer's Responses to Questions

**Comments to the Authors:**

Reviewer #1: The authors have significantly improved the manuscript. They have clarified the few issues related with the behavioral analysis methods and/or drivers’ efficiency and adequately tone down some of the conclusions. Importantly, they now better discuss their results in the context of published work. I do not have more requests on the current version.

Reviewer #2: For the most part most pressing concerns I raised the original review have not been adequately addressed:

My first major concern was the confusion regarding the authors’ conclusion of “a more deterministic function” for E-oscillators.

The authors state that they have removed the term “deterministic” from their conclusions. However, this word is still used as the concluding sentence of the abstract in the revised manuscript. And, relating to my 4th major concern (see below) it is still not clear why such a function wasn't expected of neurons previously implicated as important for late day activity.

My second major concern was over the apparent low penetrance of the cell ablation experiments. This is acknowledged by the authors in their response to the review. The survival of neurons targeted by ablation makes the results difficult to interpret cleanly.

My third major concern was regarding the lack of a Mai-179-Gal4/Pdf-GAL80/UAS-CycDN. The authors have added this to their experiments, but, as pointed out by other reviewers, the results simply confirm previous work that M cells are important for rhythms under constant darkness.

My fourth concern was that the authors did not present alternative model of clock network function based on these new results, nor did they provide a clear or compelling revision in our conception of the network or how it entrains to distinct environmental rhythms. The authors have still not sufficiently explained why the finding that the evening cells make a significant contribution to evening activity under LLTC cycles is unexpected. The authors do state that this is surprising based on previous work by Yoshii and colleagues (2010) showing that Cry- neurons have molecular clocks that are more strongly driven by temperature cycles than Cry+ neurons. They also showed that Cry- neurons were sufficient for (bimodal) rhythms under light/moonlight plus temperature cycles and that flies lacking Cry+ neurons did not re-entrain to a shifted temperature cycle when the Light-Moonlight cycle was kept at the same phase. However, it does not logically follow that this means E cells would not be expected to make significant contributions to the evening peak of activity under LLTC, if present and intact within the network. It is therefore still not clear to me why the major results are not expected. E cells drive evening activity under entrainment. Why would we have predicted otherwise?

For these reasons, I do not think my major concerns have been addressed sufficiently.

Reviewer #3: The authors have revised the manuscript and addressed most of my concerns. I have a couple comments which could help to make some points clearer.

• The following is still not clear: a) Why has synchronisation among individuals been used and is it is the same as inter-individual variation? And what can this tell us about the role of these cells under LLTC. b) The rationale of using this measure over traditional measures of anticipation index or peak timing is still unclear. c) While to some extent the calculations are now clear it is still difficult to understand why this measurement of using slope(theoretical) value was obtained.

• Why was the median used for plotting the activity profiles instead of mean? Was the data skewed. No error was reported (as previously pointed out). Since error around median does not always make sense, often range (minimum to maximum) is reported. If the slope ratio was calculated using median values what sort of statistical test was used to make the comparisons shown in Figure1 D and E. The tests have not been explicitly mentioned in the methods section.

• “Our data indicate that morning oscillator neurons dictate morning activity during DDTC (31), while evening neurons are required in LLTC for setting the activity phase to the afternoon (Fig 1).”

• An alternate view could be that under constant temperature conditions, M cells are the pacemakers in DD and E cells are the pacemakers under LL (at least when CRY is absent). Neither of these cell types are reported to express temperature sensors. Thus, temperature information must be relayed to them either from outside the clock circuit or from other clock neurons possessing temperature sensors like DN1p as mentioned in the paragraph below

**Have all data underlying the figures and results presented in the manuscript been provided?**

Reviewer #1: Yes

Reviewer #2: Yes

Reviewer #3: Yes

PLOS authors have the option to publish the peer review history of their article (what does this mean?). If published, this will include your full peer review and any attached files.

Reviewer #1: No

Reviewer #2: No

Reviewer #3: No

---

## [Decision Letter · Decision Letter 1]

7 Oct 2022

Dear Dr Stanewsky,

Thank you very much for submitting your Research Article entitled 'Light-dependent network switching between circadian morning and evening oscillators controls behavior during daily temperature cycles' to PLOS Genetics.

The manuscript was fully evaluated at the editorial level and by independent peer reviewers. You will see that the reviews leave us in a difficult situation because they make opposite recommendations. In order to help us come to a fair decision we ask that you please revise the manuscript addressing the comments of Reviewer 2. The re-revised manuscript will then be evaluated at the editorial level only. We promise a rapid turnaround.

If you decide to revise the manuscript for further consideration at PLOS Genetics, please aim to resubmit within the next 60 days, unless it will take extra time to address the concerns of the reviewers, in which case we would appreciate an expected resubmission date by email to plosgenetics@plos.org.

Please do not hesitate to contact us if you have any concerns or questions.

Yours sincerely,

John Ewer

Academic Editor

PLOS Genetics

Gregory P. Copenhaver

Editor-in-Chief

PLOS Genetics

Reviewer's Responses to Questions

**Comments to the Authors:**

Reviewer #2: The authors have added a model to the revised manuscript in response to my persistent concerns regarding how the work produces a change in our understanding of circadian timekeeping, but this model lacks sufficient mechanistic detail to be useful to the field. The authors have also replaced the original abstract that still contained the conclusion regarding the work supporting a "more deterministic" function for E oscillators. These are improvements. But my concerns remain, notwithstanding the elaborate discussion of previous work in the authors' response to the review. The results described in the manuscript largely support previous work indicating that the balance of control shifts within the clock neuron circuitry as a function of environmental conditions. This manuscript covers familiar ground in this regard: M cells dominate in darkness and non M cells dominate in the light. I am still not convinced that the manuscript significantly changes our conception of the network in a clear or compelling way.

Reviewer #3: The authors have addressed my concerns and provided their rationale in the response to reviewers file.

**Have all data underlying the figures and results presented in the manuscript been provided?**

Reviewer #2: Yes

Reviewer #3: None

PLOS authors have the option to publish the peer review history of their article (what does this mean?). If published, this will include your full peer review and any attached files.

Reviewer #2: No

Reviewer #3: No

---

## [Editor Report · Decision Letter 2]

20 Oct 2022

Dear Dr Stanewsky,

We are pleased to inform you that your manuscript entitled "Light-dependent network switching between circadian morning and evening oscillators controls behavior during daily temperature cycles" has been editorially accepted for publication in PLOS Genetics. Congratulations! The only suggestion we would make is that you consider changing the title so it more clearly conveys the idea that the work explores how light and temperature are integrated by the clock.

Yours sincerely,

John Ewer

Academic Editor

PLOS Genetics

Gregory P. Copenhaver

Editor-in-Chief

PLOS Genetics

Comments from the reviewers (if applicable):

**Data Deposition**

http://datadryad.org/submit?journalID=pgenetics&manu=PGENETICS-D-22-00633R2

**Press Queries**

---

## [Editor Report · Acceptance letter]

7 Nov 2022

PGENETICS-D-22-00633R2 

Light-dependent network switching between circadian morning and evening oscillators controls behavior during daily temperature cycles 

Dear Dr Stanewsky, 

We are pleased to inform you that your manuscript entitled "Light-dependent network switching between circadian morning and evening oscillators controls behavior during daily temperature cycles" has been formally accepted for publication in PLOS Genetics! Your manuscript is now with our production department and you will be notified of the publication date in due course.

With kind regards,

Zsofia Freund

PLOS Genetics

On behalf of:
